# Using Deep Convolutional Neural Networks for Enhanced Ultrasonographic Image Diagnosis of Differentiated Thyroid Cancer

**DOI:** 10.3390/biomedicines9121771

**Published:** 2021-11-26

**Authors:** Wai-Kin Chan, Jui-Hung Sun, Miaw-Jene Liou, Yan-Rong Li, Wei-Yu Chou, Feng-Hsuan Liu, Szu-Tah Chen, Syu-Jyun Peng

**Affiliations:** 1Division of Endocrinology and Metabolism, Department of Internal Medicine, Chang Gung Memorial Hospital, College of Medicine, Chang Gung University, Taoyuan 33302, Taiwan; wkchanmd@gmail.com (W.-K.C.); ma1233@cgmh.org.tw (J.-H.S.); lioumj@cgmh.org.tw (M.-J.L.); mr8252@cgmh.org.tw (Y.-R.L.); weiyu@chhome.net (W.-Y.C.); fenghsuanliu@gmail.com (F.-H.L.); stc1105@cgmh.org.tw (S.-T.C.); 2Professional Master Program in Artificial Intelligence in Medicine, College of Medicine, Taipei Medical University, Taipei 10675, Taiwan

**Keywords:** thyroid cancer, artificial intelligence, deep learning, CNNs

## Abstract

Differentiated thyroid cancer (DTC) from follicular epithelial cells is the most common form of thyroid cancer. Beyond the common papillary thyroid carcinoma (PTC), there are a number of rare but difficult-to-diagnose pathological classifications, such as follicular thyroid carcinoma (FTC). We employed deep convolutional neural networks (CNNs) to facilitate the clinical diagnosis of differentiated thyroid cancers. An image dataset with thyroid ultrasound images of 421 DTCs and 391 benign patients was collected. Three CNNs (InceptionV3, ResNet101, and VGG19) were retrained and tested after undergoing transfer learning to classify malignant and benign thyroid tumors. The enrolled cases were classified as PTC, FTC, follicular variant of PTC (FVPTC), Hürthle cell carcinoma (HCC), or benign. The accuracy of the CNNs was as follows: InceptionV3 (76.5%), ResNet101 (77.6%), and VGG19 (76.1%). The sensitivity was as follows: InceptionV3 (83.7%), ResNet101 (72.5%), and VGG19 (66.2%). The specificity was as follows: InceptionV3 (83.7%), ResNet101 (81.4%), and VGG19 (76.9%). The area under the curve was as follows: Incep-tionV3 (0.82), ResNet101 (0.83), and VGG19 (0.83). A comparison between performance of physicians and CNNs was assessed and showed significantly better outcomes in the latter. Our results demonstrate that retrained deep CNNs can enhance diagnostic accuracy in most DTCs, including follicular cancers.

## 1. Introduction

Most thyroid tumors are incidentally discovered via palpation by clinical physicians. It has been estimated that the prevalence of thyroid cancer can reach 65% [1] and is more common among females. Fortunately, most tumors are benign thyroid nodules; that is, only a small number of them are malignant [2]. Roughly 5–10% of these tumors are identified as thyroid cancer. In Taiwan, thyroid cancer is becoming increasingly common with most cases identified in individuals between 40 and 65 years old, and it is currently the fourth most prevalent form of cancer among women, as well as the most common cancer of the endocrine system. The Health Promotion Administration of Taiwan has reported a 9.67% annual increase in the number of newly diagnosed cases of thyroid cancer. This may be due in part to advances in ultrasound and imaging technology over the past decade, which have greatly facilitated diagnostic procedures [3], particularly when dealing with tumors measuring less than 1 cm. The prognosis in cases of thyroid cancer is generally good. At present, the long-term prognosis after standard treatment for differentiated thyroid cancers (DTC) is excellent, with a 10 year survival rate of 96% [4].

Risk factors of thyroid cancer include exposure to radiation in one’s youth, a history of thyroid goiter, and a family history of cancer [5]. The histopathology of thyroid cancers can be classified according to the state of follicular epithelial cells as DTC, poorly differentiated thyroid carcinoma, and anaplastic thyroid carcinoma (ATC) [6], as well as medullary thyroid cancer (MTC) derived from C cells [7]. DTC is the most common form of thyroid cancer, which appears as papillary thyroid carcinoma (PTC) in 90–92% of cases and the follicular variant of papillary thyroid carcinoma (FVPTC; the most common subtype of PTC). Rare forms include follicular thyroid carcinoma (FTC) and Hürthle cell carcinoma (HCC), both of which have proven difficult to diagnose [8]. At present, the prognosis for rare variants other than differentiated thyroid cancer is very poor, due to a lack of effective treatment options [6].

Currently, the gold standard for the clinical diagnosis of thyroid cancers is ultrasound-guided fine-needle aspiration or core-needle biopsy combining cytological and pathological analysis [9]. Unfortunately, this method is applicable only to PTC. Identifying other types of DTC (FVPTC, FTC, and HTC) is hampered by a lack of distinguishing cytological characteristics in ultrasound images (Figure 1), such as hypo-echogenicity, irregular margins, or microcalcifications [10]. In many cases, surgical resection of the tumor is necessary to confirm DTC subtypes, such as FTC and HCC. Molecular biology and genetic analysis can be used to facilitate diagnosis [11]; however, the tools and expertise required for such analysis are generally available only in medical centers. Note that the invasive nature of fine-needle aspiration and core-needle biopsy inevitably leads to complications, such as bleeding or infection. Furthermore, the effectiveness of the procedures depends largely on the experience and skill of the operator [12].

Artificial intelligence (AI) is a key technology in the on-going personalization and development of precision medicine. Musko [13] claimed that artificial intelligence (AI) allows doctors and researchers to make predictions of greater accuracy, thereby making it easier to identify the treatment and prevention strategies best suited to a particular disease and/or groups of patients. Deep learning algorithms are increasingly being used to facilitate the diagnosis of tumors. Chi [14] reported that the retrained GoogleNet outperformed conventional machine learning approaches, such as support vector machine (SVM). Using InceptionV3, Song [15] achieved diagnostic performance comparable to that of experienced professional radiologists. Various deep learning models have also been trained to differentiate between malignant and benign thyroid tumors [16,17,18,19,20,21,22,23]. A number of studies have addressed the issue of training AI systems in the analysis of thyroid ultrasound images; however, most of this work has focused on PTC. Diagnosing other pathological types (FVPTC, FTC, and HCC) is hindered by their rarity in clinical practice and their similarity to benign lesions in ultrasound images. The prognosis for DTCs should be similar to that of PTC as long as they are identified early. Ideally, clinicians should be able to confirm diagnosis prior to surgical intervention.

In this study, transfer learning was used to train a deep convolutional neural network (CNN) [24] for the analysis of ultrasound images with the aim of differentiating between malignant and benign thyroid lesions, and facilitating the identification of other DTCs (e.g., FTC). We anticipate that such CNN models could help to eliminate unnecessary invasive examinations or surgical interventions.

## 2. Materials and Methods

### 2.1. Data Sources

This retrospective study was based on medical records and clinical data from the cancer registry of Chang Gung Memorial Hospital (Linko branch) covering the period from January 2008 to July 2020. Patients aged >20 years with surgically confirmed diagnosis of thyroid cancer were enrolled. Furthermore, patients with surgically confirmed benign thyroid nodules between January 2016 and July 2020 were also enrolled. Patients with non-DTC were excluded (Figure 2). Patients who had not undergone an ultrasound examination within 12 months prior to surgical intervention were also excluded, as were patients without recognizable cancer lesions in ultrasound images. This study was approved by the Institutional Review Board of the Chang Gung Medical Foundation (IRB No. 202001440B0, 31 August 2020). The requirement for informed consent was waived due to the retrospective nature of this analysis.

### 2.2. Data Collection

Demographic and clinical data included the age of the patient at the time of diagnosis, gender, lesion location (left, right, both, or isthmus), ultrasound manufacturer (e.g., Aloka, Hitachi, and Siemens) (Table 1), the distribution of pathological groups as a function of ultrasound brand (Table 2), and histopathological data (Table 3). All images were downloaded and stored in TIFF format. Every patient included in the study presented at least one thyroid tumor in ultrasound images (longitudinal or horizontal view) when assessed using the models of multiple ultrasound manufacturers. An ultrasound image may be formed by a nodule in two views as long as they were saved in double-view mode. After a manual review of the examination data, researchers collected 1791 ultrasound images for analysis. Note that the regions of interest (ROIs) in the ultrasound images were marked by the author as rectangle bounding boxes (Figure 3). The ROI was meant to include the entire tumor except in cases where the tumor exceeded the image boundary, such that the bounding box included only the visible part of the tumor. Ultrasound images were subsequently cropped according to the bounding boxes, which resulted in 2308 images of nodules for training (Figure 4). The images were divided into a training set (80%) and a test set (20%). Images from each patient were placed in either the training set or the test set, but not in both. Image data underwent preprocessing to compensate for the relatively small number of images and reduce the likelihood of overfitting. As shown in Figure 5, data augmentation based on histogram equalization/normalization and horizontal flipping increased the number of images by four times, as follows: original image, image with histogram equalization, image with horizontal flipping, and images with histogram equalization and horizontal flipping. The training set included 3316 images showing malignant tumors and 4044 images showing benign tumors. All images showing Hürthle cell adenoma (HA) were included in the training set. The test set (used to assess diagnostic performance) included 204 images showing malignant tumors and 264 images showing benign tumors.

### 2.3. Study Design

The diagnoses of all tumors in this study were subject to surgical and pathological confirmation; therefore, training was implemented as supervised learning. Transfer learning and fine-tuning of hyperparameters were implemented on three pretrained CNNs, namely, InceptionV3, ResNet101, and VGG19. Note that the classification accuracy of these CNNs has been demonstrated in the ImageNet Large-Scale Visual Recognition Challenge (ILSVRC). The MATLAB 2021a platform was used for the retraining of the three CNNs to classify benign and malignant thyroid tumors in ultrasound images. The size of the input images was adjusted according to the CNN settings. Stochastic gradient descent with momentum (SGDM) was applied as the solver. The maximum epochs were as follows: InceptionV3 (26), ResNet101 (21), and VGG19 (32). The learning rate was as follows: InceptionV3 (0.001), ResNet101 (0.001), and VGG19 (0.0001). Fivefold cross-validation was used to ensure the stability of the results.

### 2.4. Statistical Analysis

This study compared the diagnostic capability of CNNs with that of two endocrinologists with over 20 years of experience in performing fine-needle aspiration and the interpretation of ultrasound images on the test set. In estimating the diagnosis performance of physicians, the images were classified as malignant and benign according to sonographic patterns and estimated risk of malignancy, as suggested in the American Thyroid Association (ATA) classification system [25]. The CNNs were assessed in terms of accuracy, sensitivity, specificity, positive predictive value (PPV), and negative predictive value (NPV), as well as the receiver operating characteristic (ROC) curve, area under the curve (AUC), and confusion matrix. We also assessed accuracy in identifying tumors with various histopathologies. Continuous variables were presented as the mean and standard deviation (SD), as indicated. Categorical data were expressed in terms of actual frequencies and percentages. Statistical analysis was performed using the chi-square test and analysis of variance (ANOVA). *p*-Values < 0.05 were considered significant. All statistical analysis was conducted using the SAS Suite, version 9.4 (SAS Institute, Cary, NC, USA).

## 3. Results

### 3.1. Study Population

A total of 791 patients were identified in our initial analysis (Figure 1). From this group, 17 patients were excluded due to non-DTC, including anaplastic cancer (*n* = 6), medullary cancer (*n* = 7), and metastatic cancer (*n* = 4). Patients who had not undergone ultrasound examinations within 12 months prior to surgery were also excluded, as were those without recognizable lesions (*n* = 353). This left 421 DTC patients and 391 patients with benign thyroid nodules who met the enrollment criteria for this study.

### 3.2. Demographics

As shown in Table 1, the patients were divided into a malignant group (comprising a PTC group, FTC group, FVPTC group, and HCC group) and a benign group. The mean age of patients was 44.9–54.2 years old. The average age at the time of diagnosis was lower in the malignant groups (*p* < 0.0001). We observed a higher proportion of females in all groups; however, the female-to-male ratio between groups did not differ significantly. We observed statistically significant between-group differences in terms of lesion location (*p* = 0.0017) with very few instances of bilateral lesion. In malignant groups, the PTC group presented the highest proportion of simultaneous bilateral lesions (5.14%), whereas the FTC group presented the highest proportion of isthmus lesions (4.29%). We observed statistically significant between-group differences in terms of ultrasound manufacturer (*p* < 0.0001). The most common brands in the PTC group were GE Healthcare (37.12%) and Siemens (28.82%), and the most common brand in the other groups was GE Healthcare. Table 2 lists the distribution (percentages) of pathological groups as a function of ultrasound brand. Statistically significant differences were observed between all pathological groups as a function of ultrasound brand (*p* < 0.001).

Table 3 lists the histopathological distribution of tumors among groups. On the basis of ultrasonic features, the malignant group was divided into PTC and FTC subgroups. The PTC subgroup included classic PTC, the diffuse sclerosing variant, the tall cell variant, the cribriform morular variant, and the encapsulated variant. The FTC sub-group included FTC, FVPTC, HCC, and the encapsulated follicular variant of PTC. The benign group included nodular hyperplasia (NH), follicular adenoma (FA), cysts, and HA. Classic PTC and FVPTC were the most common pathology types in the PTC and FTC subgroups, respectively. Nodular hyperplasia was the most common feature in the benign group.

### 3.3. Performance Assessment on CNNs and Physician

The training set contained a total of 7360 nodule images after data augmentation, including 1744 in the PTC group, 852 in the FVPTC group, 568 in the FTC group, 152 in the HCC group, and 4044 in the benign group. Following the completion of transfer learning, the test set was used to assess the performance of the CNNs and obtain a confusion matrix, as shown in Figure 6. Table 4 presents the accuracy of the CNNs and physicians in terms of diagnostic performance, as follows: InceptionV3 (76.5%), ResNet101 (77.6%), VGG19 (76.1%), Endocrinologist 1 (58.8%), and Endocrinologist 2 (62%). The sensitivity was as follows: InceptionV3 (83.7%), ResNet101 (72.5%), VGG19 (66.2%), Endocrinologist 1 (38.7%), and Endocrinologist 2 (35.3%). The specificity was as follows: InceptionV3 (83.7%), ResNet101 (81.4%), VGG19 (76.9%), Endocrinologist 1 (72.4%), and Endocrinologist 2 (82.6%). A confusion matrix illustrating multiclass classification using ResNet101 in the test set is shown in Appendix A. Due to the small number of cases in the malignant groups (e.g., FVPTC, FTC, and HCC), the accuracy of the CNN was only 65%.

Table 5 lists the accuracy of the CNNs and physicians in identifying tumors with various pathological types. In the identification of malignant tumors, the highest accuracy in diagnosing PTC was 81.4% (InceptionV3), the highest accuracy in diagnosing FVPTC was 74.6% (ResNet101), the highest accuracy in diagnosing FTC was 72.7% (InceptionV3), and the highest accuracy in diagnosing HCC was 66.7% (InceptionV3 and ResNet101). In identifying benign tumors, the highest accuracy in diagnosing NH was 82.4% (VGG19), the highest accuracy in diagnosing FA was 80% (ResNet101), and the highest accuracy in diagnosing cysts was 95% (VGG19). In terms of the performance of physician diagnosis, Endocrinologist 1 showed better accuracy in diagnosing PTC (58.8% vs. 53.6%), FVPTC (20.3% vs. 17%), HCC (13.3% vs. 6.7%), and FA (80% vs. 75%), while Endocrinologist 2 showed higher accuracy in FTC (30.3% vs. 27.3%), NH (81.9% vs. 73%), and cyst (90% vs. 80%). Figure 7 presents the ROC curve of CNNs and performance of physicians. As shown in Table 3, the area under the curve (AUC) was as follows: InceptionV3 (0.82), ResNet101 (0.83), and VGG19 (0.83).

## 4. Discussion

In this study, deep convolutional neural networks (CNNs) were used to classify thyroid tumors as malignant or benign. Note that the accuracy achieved in the current study was slightly lower than in previous studies [15,16,17,21,22]. To the best of our knowledge, this was the first study focusing on the use of CNNs for the classification of DTCs other than PTC (FVPTC, FTC, and HCC). Diagnosing malignant thyroid tumors (e.g., FTC) prior to surgical intervention remains an unresolved problem [26]. A definitive diagnosis of FTC requires surgical intervention. This issue is largely due to similarities between the ultrasonic features of malignant and benign nodules [27]. From a microscopic point of view, the major difference between malignant FTC and benign follicular tumors (FAs) is the occurrence of vascular or capsular invasion. Furthermore, cytopathological analysis is often inconclusive due to a lack of distinguishing characteristics, such as the “clear cell border” and “pseudo-inclusion body” observed in PTC cells. Note that most of these cases would eventually be classified as follicular neoplasms (FNs) [28]. Technical advances in molecular biology and genetic engineering have revealed a link between BRAF mutations and PTC, as well as a link between RAS mutations and FTC [11]. Unfortunately, the cost of molecular and genetic testing is prohibitive in most cases and unavailable except in the best-equipped medical centers. Due to the low incidence of malignant thyroid tumor, the inclusion of advanced diagnostics in routine thyroid examinations is unreasonable. More importantly, diagnostic accuracy is strongly influenced by the number of successful punctures in fine-needle aspiration.

In one recent study, machine learning methods proved highly effective in diagnosing FNs [29]. In fact, a thyroid CAD named AmCAD-UT has already been approved by the United States Food and Drug Administration (FDA) and Taiwan Medical Device Marketing Approval for the assessment of thyroid tumors using feature extraction/selection. AI is proving highly effective in overcoming the difficulties associated with the diagnosis of malignant tumors.

The ImageNet project has been instrumental in advancing computer vision and deep learning research. ImageNet provides an image database based on the WordNet hierarchy, and data are freely available to researchers for noncommercial applications [30]. The database has been manually annotated with more than 14 million images. Between 2010 and 2017, ImageNet held an annual competition (referred to as ILSVRC) to evaluate algorithms used in object detection and image classification. The CNNs used in this study for transfer learning achieved the highest classification accuracy in 2014 (Inception), the highest detection results in 2014 (VGG), and the highest classification accuracy in 2015 (ResNet). Since 2015, the accuracy of deep learning image classification has exceeded 95%, far exceeding the capabilities of humans. In the intervening years, new CNNs (e.g., SEnet) have achieved even higher accuracy; however, the difference is negligible. Most of the previous thyroid cancer imaging studies using Inception, ResNet, and VGG achieved acceptable accuracy and were, therefore, deemed suitable for transfer learning in the current study. We discovered that the less complex CNNs (Inception and ResNet) were slightly faster than VGG in terms of training and classification; however, overall classification accuracy was nearly identical.

In identifying cases of PTC, we were unable to achieve the 90% accuracy observed in previous studies [17,22], due primarily to an insufficient number of cases. Note that most of the previous CNN studies on classic PTC included at least 1000 patients. According to the results in previous studies, it appears likely that accuracy in identifying PTC could be increased simply by including a larger number of cases. It is also possible that the performance of the CNN algorithms was hindered by unacceptably low image resolution after cropping. There is also a possibility that the apparatus used for ultrasonic imaging played a role in algorithm performance, due to subtle differences in image fineness, brightness, contrast, and texture output from different ultrasound manufacturers.

In our analysis, accuracy in identifying FVPTC reached 74.6% (ResNet101), which is similar to the results obtained for classic PTC. FVPTC is the most common PTC variant and the second largest group in the current study. The ultrasonic characteristics of FVPTC differ considerably from those of classic PTC and, in many respects, are similar to those of benign tumors [31]. Our results demonstrated that accuracy in identifying FTC was only 63.6–72.7% and accuracy in identifying FA was only 65–80%, regardless of the CNN. Accuracy in identifying HCC was only 60–66.7%, due largely to the small number of cases in the database. The CNNs seemed not to provide much benefit in the identification or diagnosis of FTC or HCC, due largely to a lack of cases resulting from low incidence and prevalence. Note that there is only a slight difference between FTC, HCC, and FA in terms of gross structure and ultrasound features [32,33].

The performance of classification by retrained CNNs was a lot better than that of the participating physicians, especially in the malignant groups. The poor diagnostic performance of physicians in dealing with malignant tumors resulted in poor sensitivity. In clinical practice, the endocrinologist or radiologist usually considers malignant features suggested by the image as a whole, not just the cropped area surrounding the tumor with low resolution. However, with the help of fine-needle aspiration and cytopathological analysis, the sensitivity of physicians may be comparable to CNNs retrained by ultrasound images alone. Remarkably low accuracy in the classification of malignant tumors by physicians also indicates the difficulty in clinical diagnosis, particularly in cases of FTC, FVPTC, and HCC. Overall, it was demonstrated that the diagnostic performance of the CNNs exceeded that of the physicians.

Overall, InceptionV3 achieved the highest sensitivity, whereas ResNet101 and VGG19 achieved higher specificity. The concurrent application of all three CNNs appears to be a viable possibility. InceptionV3 could be used to confirm a diagnosis of malignancy, whereas ResNet101 and VGG19 could be used to confirm that lesions are indeed benign.

There are numerous situations in which CAD could advantageously be implemented in conjunction with AI. For example, many developing countries lack the medical resources, professional radiologists, and endocrinologists required to obtain a reliable diagnosis of thyroid lesions. CAD could be used to screen for potential thyroid cancers for referral to a medical center. Even in medical centers, CAD could be used to facilitate the training of medical students and inexperienced physicians. More importantly, CAD could provide helpful advice in dilemmatic cases with inconclusive cytology results. From the perspective of healthcare and therapeutics, AI has also been shown to play an important role in treatment quality. Fionda [34] reported that the use of AI-based predictive models and decision support systems for radiation oncology and interventional radiotherapy can alleviate many time-consuming repetitive tasks, thereby enabling a corresponding decrease in healthcare costs.

In the current study, image ROIs were manually cropped by the author using a bounding box. This method is precise but time-consuming, and different physicians would no doubt differ in their approach to cropping. Deep learning models with auto-detection or auto-segmentation (e.g., YOLOV3 and R-CNN) could be developed to increase the speed of ROI framing before undergoing a manual review and adjustment. In the clinical application of CAD, it is also necessary to establish a graphical user-interface (GUI) capable of automating the process of ROI framing and classification.

This study was subject to a number of limitations. Firstly, the retrospective design of this study made selection bias inevitable. Secondly, an extended acquisition period was required to obtain a usable number of samples in the malignant groups. Thus, it was inevitable that the collection period for malignant cases would far exceed that of the benign control group. Thirdly, the small number of cases in our test set may have also had a negative effect on accuracy. Fourthly, the process of image selection was complicated and slow. Unlike computed tomography and magnetic resonance imaging, ultrasound images do not present a unified arrangement and require extensive manual preprocessing. In this study, the author had to review all of the ultrasound images in a search for the target lesions identified surgically. Ultrasound images also tend to vary considerably in terms of size and zooming ratio. This made it impossible to measure the tumor sizes retrospectively, leading to instances of missing data and/or mismatch with pathology reports. Note that this was the reason for the omission of tumor size in this study. Fifthly, we opted not to include the rarer forms of malignant tumor, such as undifferentiated thyroid cancers and metastatic cancers. Note that applying the current CAD in clinical practice would no doubt raise concerns about missed diagnoses. Sixthly, most of the images selected for CNN training presented identifiable single nodules. Thus, the diagnostic power in dealing with multinodular goiters with ill-defined margins remains unclear. Lastly, differences in the output algorithms of ultrasound machines can have a profound effect on training and classification. However, without raw data, there is no simple way to standardize images. Thus, the only viable approach to balancing the datasets is to apply histogram equalization or collect more images from different ultrasound manufacturers.

## 5. Conclusions

Advanced deep CNN models that are fine-tuned using transfer learning show considerable potential as a noninvasive approach to the diagnosis of DTCs, including FTC. Clinicians should be able to diagnose thyroid cancer more easily by combining ultrasound with CAD. Anticipated advances in ultrasound technology and larger databases will greatly enhance the efficacy of these methods.

## Figures and Tables

**Figure 1 biomedicines-09-01771-f001:**
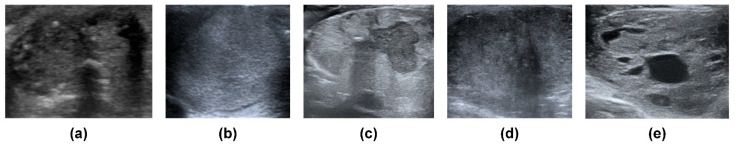
Ultrasound images of (**a**) papillary thyroid carcinoma, (**b**) follicular variant of papillary thyroid carcinoma, (**c**) follicular thyroid carcinoma, (**d**) Hürthle cell carcinoma, and (**e**) benign thyroid nodule.

**Figure 2 biomedicines-09-01771-f002:**
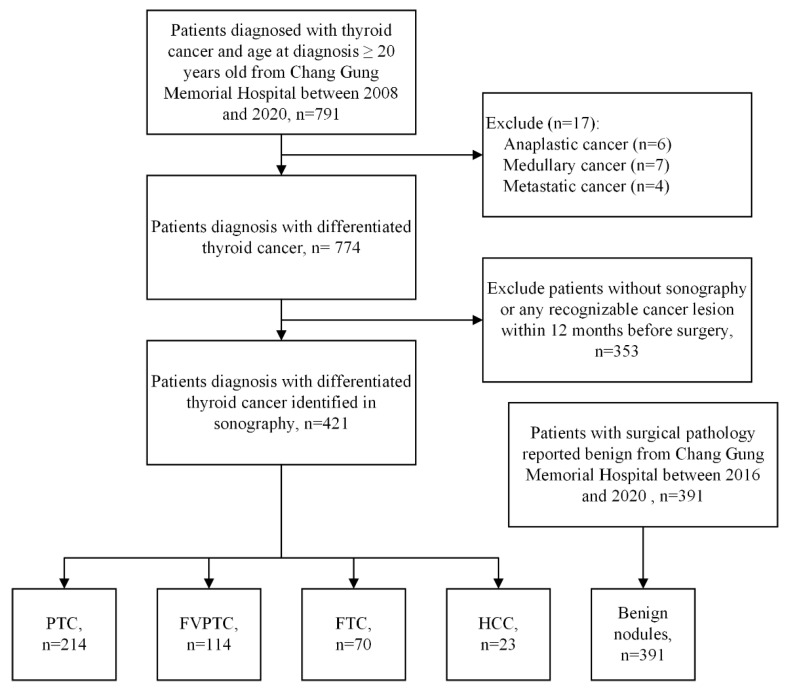
Flowchart of study population. A total of 421 differentiated thyroid cancer patients and 391 patients with benign nodules enrolled after excluding other cancer types and patients without recognizable lesion in sonography. Papillary thyroid carcinoma (PTC), follicular thyroid carcinoma (FTC), follicular variant of papillary thyroid carcinoma (FVPTC), Hürthle cell carcinoma (HCC).

**Figure 3 biomedicines-09-01771-f003:**
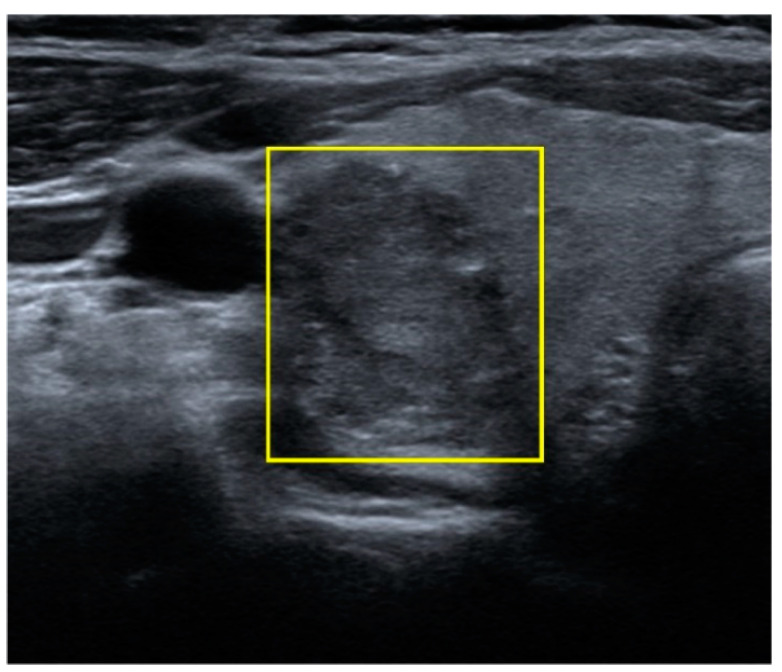
Ultrasound image of PTC. ROI cropped by a rectangle frame drawn by the author. Region of interest (ROI), papillary thyroid carcinoma (PTC).

**Figure 4 biomedicines-09-01771-f004:**
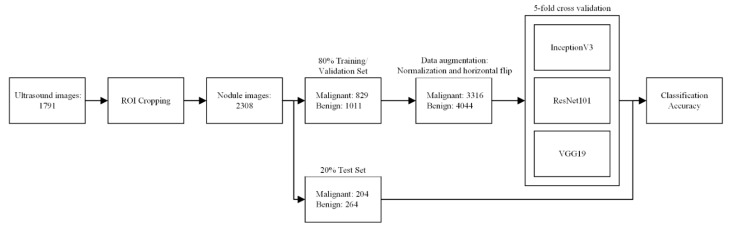
Process of training and classification. After preprocessing and ROI cropping of nodule images, 2308 images were split into a training set and test set. Data augmentation based on histogram equalization/normalization and horizontal flipping increased the number of images fourfold. Transferred learning by three pretrained CNNs was performed with fivefold cross-validation; then, the diagnostic performance using the test set was evaluated. Region of interest (ROI), convolution neural network (CNN).

**Figure 5 biomedicines-09-01771-f005:**
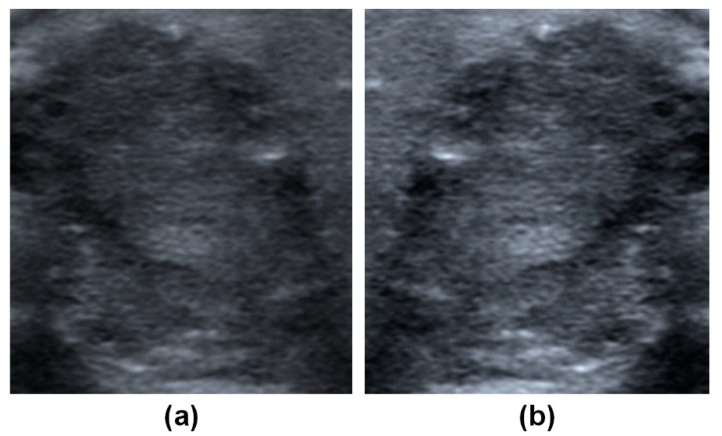
(**a**) Cropped image of PTC nodule. (**b**) Cropped image was preprocessed with horizontal flipping and histogram normalization. Papillary thyroid carcinoma (PTC).

**Figure 6 biomedicines-09-01771-f006:**
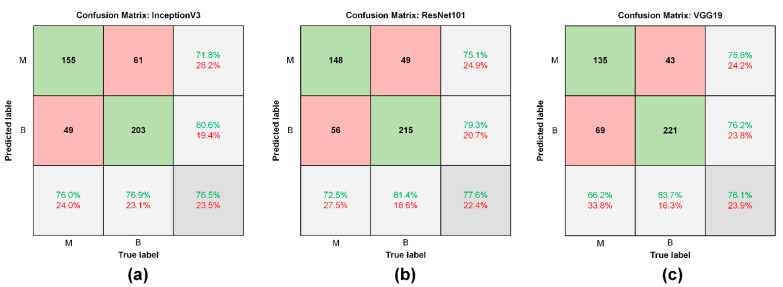
Confusion matrix of CNNs in test set. (**a**) InceptionV3, (**b**) ResNet101, and (**c**) VGG19. M: malignant group, B: benign group.

**Figure 7 biomedicines-09-01771-f007:**
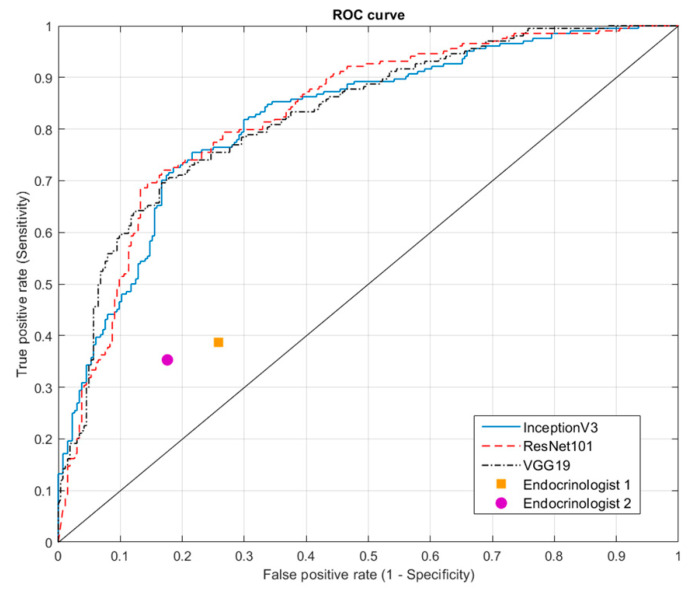
ROC curve of CNN models and performance of endocrinologists. AUC of InceptionV3 = 0.81, AUC of ResNet101 = 0.81, and AUC of VGG19 = 0.80. Receiver operating characteristic (ROC), convolution neural network (CNN), area under the curve (AUC).

**Table 1 biomedicines-09-01771-t001:** Baseline demography of five pathology types divided into malignant and benign groups.

	Pathological Types					
	Malignant Group (*n* = 421)				Benign Group (*n* = 391)	
	PTC	FVPTC	FTC	HCC	Benign	*p*-Value
Number of patients	214	114	70	23	391	
Age (years), mean (SD)	47.36 ± 13.70	44.90 ± 15.12	46.61 ± 17.10	51.17 ± 15.62	54.18 ± 13.15	<0.0001
Sex (*n*, %)						
Male	53 (25)	28 (25)	15 (21)	4 (17)	80 (20)	0.6985
Female	161 (75)	86 (75)	55 (79)	19 (83)	311 (80)	
Number of US images	470	215	131	38	937	
Number of cropped images	533	272	175	53	1275	
Location (*n*, %)						
Left	88 (41.12)	52 (45.61)	39 (55.71)	13 (56.52)	143 (36.57)	0.0017
Right	109 (50.93)	57 (50.00)	27 (38.57)	10 (43.48)	192 (49.10)	
Both (Left + Right)	11 (5.15)	4 (3.51)	1 (1.43)	0 (0.00)	47 (12.03)	
Isthmus	6 (2.80)	1 (0.88)	3 (4.29)	0 (0.00)	9 (2.30)	
Ultrasound brands (%)						
Aloka	3.93	2.46	6.67	0.00	7.57	<0.0001
GE Healthcare	37.12	62.30	64.01	78.26	45.87	
Hitachi	7.42	2.46	1.33	0.00	7.34	
Philips	4.37	0.82	8.00	8.70	2.06	
Siemens	28.82	10.66	5.33	8.70	18.81	
Toshiba	17.90	16.39	13.33	4.34	18.12	
Others	0.44	4.91	1.33	0.00	0.23	

Continuous data are expressed as the mean and standard deviation (SD); categorical data are expressed as a percentage (%). Papillary thyroid carcinoma (PTC), follicular thyroid carcinoma (FTC), follicular variant of papillary thyroid carcinoma (FVPTC), Hürthle cell carcinoma (HCC), ultrasound (US).

**Table 2 biomedicines-09-01771-t002:** Distribution of pathological groups as a function of ultrasound brands.

	Aloka	GE Healthcare	Hitachi	Philips	Siemens	Toshiba	Others	*p*-Value
PTC	18.00	19.90	32.07	35.72	39.52	27.15	11.11	<0.0001
FVPTC	6.00	17.80	5.66	3.57	7.78	13.25	66.67	<0.0001
FTC	10.00	11.24	1.89	21.43	2.40	6.62	11.11	<0.0001
HCC	0.00	4.22	0.00	7.14	1.20	0.66	0.00	<0.0001
Benign	66.00	46.84	60.38	32.14	49.10	52.32	11.11	<0.0001

Categorical data are expressed as a percentage (%). Papillary thyroid carcinoma (PTC), follicular thyroid carcinoma (FTC), follicular variant of papillary thyroid carcinoma (FVPTC), Hürthle cell carcinoma (HCC).

**Table 3 biomedicines-09-01771-t003:** Distribution of histopathological and ultrasonic features among groups.

		Histopathology	Number (%)
Malignant group (*n* = 421)	PTC features (*n* = 214)	Classic PTC	208 (97.19)
		Diffuse sclerosing variant	3 (1.40)
		Tall cell variant	1 (0.47)
		Cribriform morular variant	1 (0.47)
		Encapsulated variant	1 (0.47)
	FTC features (*n* = 207)	Follicular variant of PTC	106 (51.21)
		Follicular carcinoma, minimally invasive	70 (33.82)
		Hürthle cell carcinoma	23 (11.11)
		Encapsulated follicular variant of PTC	8 (3.86)
Benign group (*n* = 391)		Nodular hyperplasia	289 (73.91)
		Follicular adenoma	48 (12.28)
		Cyst	47 (12.02)
		Hürthle cell adenoma	7 (1.79)

Papillary thyroid carcinoma (PTC), follicular thyroid carcinoma (FTC).

**Table 4 biomedicines-09-01771-t004:** Performance of CNNs and endocrinologists in test set.

	Sensitivity	Specificity	PPV	NPV	Accuracy	AUC
InceptionV3	76.0	76.9	71.8	80.6	76.5	0.82
ResNet101	72.5	81.4	75.1	79.3	77.6	0.83
VGG19	66.2	83.7	75.8	76.2	76.1	0.83
Endocrinologist 1	38.7	74.2	53.7	61.1	58.8	-
Endocrinologist 2	35.3	82.6	61.0	62.3	62.0	-

Positive predictive value (PPV), negative predictive value (NPV), area under curve (AUC).

**Table 5 biomedicines-09-01771-t005:** Diagnostic accuracy of CNNs and endocrinologists on different pathological types in test set.

	Malignant Group				Benign Group		
	PTC	FVPTC	FTC	HCC	NH	FA	C
InceptionV3	81.4	72.9	72.7	66.7	75.0	65.0	92.5
ResNet101	73.2	74.6	69.7	66.7	79.4	80.0	90.0
VGG19	64.9	71.2	63.6	60.0	82.4	75.0	95.0
Endocrinologist 1	58.8	20.3	27.3	13.3	73.0	80.0	80.0
Endocrinologist 2	53.6	17.0	30.3	6.7	81.9	75.0	90.0

Papillary thyroid carcinoma (PTC), follicular thyroid carcinoma (FTC), follicular variant of papillary thyroid carcinoma (FVPTC), Hürthle cell carcinoma (HCC), Nodular hyperplasia (NH), Follicular adenoma (FA), Cyst (C).

## Data Availability

Data sharing is not applicable to this article.

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
