# Peer review of "Using Deep Convolutional Neural Networks for Enhanced Ultrasonographic Image Diagnosis of Differentiated Thyroid Cancer"

_biomedicines, 2021, doi:10.3390/biomedicines9121771_

Round 1
Reviewer 1 Report
The authors present a system that uses convolutional neural networks (CNN) to distinguish benign lesions thyroid cancer using ultrasound images. Thyroid cancer is further subdivided in four categories. The system is trained with 421 thyroid cancer patients and 391 benign controls. From such cases, the authors collect 2308 images of nodules, found by placing manual ROIs. The images are split into training (80%) and test (20%). Standard CNNs are trained, starting from the weights obtained from natural images. Data augmentation based on histogram equalization and flipping is performed. The test set included 204 images of malignant tumors and 264 images of benign lesions. The performance of the CNNs is high in all measurements, outperforming endocrinologists in all of them except specificity.
The rationale and justification of the proposed work is correct. However, I have serious concerns with respect to the materials and methods. Overall, further data analysis should be performed to demonstrate that the CNNs are truly detecting benign vs. malignant lesions instead of image properties that are subtle to the acquisition. CNNS are well known to suffer from overfitting to image properties instead than to the subject of interest.
More precisely, the authors should discuss and investigate if there is a bias in the cohort selection. The cohorts are acquired from different periods of time. It is unclear the cancer cohort acquisition period. In line 90 it is stated that cancers cases are obtained in the period 2003-2020 (line 90) or 2008-2020 (Figure 1). Please reconcile such data. Controls are obtained from 2016-2020. It is unlikely that an image acquired in 2003 has the same properties as an image of 2020. At least an analysis on the cancer cases between 2016-2020 should be performed.
The authors claim in the data split that it is 80%-20% for training and testing. However, did the authors perform such split in such a way that no images from the same case appear in the training and testing sets? Images from the same subject should be either in training or testing, but not in both. Please clarify.
Similarly, the authors provide a split by acquisition device brand. Within each brand, are they always the same device? Since the study period is large (17 years), it is unlikely that the same devices are used throughout the study.
In Table 1, the percentages of ultrasound brands would be more informative if they are read as, for instance, percentage of PTC measured with Aloka devices from all cases measured with Aloka devices. This would let the reader know potential biases by device.
A confusion matrix of the multi-class classification should be included since it would be very informative. One network would suffice.
Images representative of all tumor types should be included to provide the reader with an intuition of the depth and difficulties of the problem.
Due to the above mentioned issues with data handling I recommend a major review of the manuscript.
Reviewer 2 Report
The authors present a paper about "Using Deep Convolutional Neural Networks for Enhanced Ultrasonographic Image Diagnosis of Differentiated Thyroid Cancer".
The topic is extremely interesting because Artificial Intelligence is becoming more and more important in clinical practice.
The article is overall well written but I have some minor suggestions to improve the paper as follows:
1) In the introduction section it could be useful to underline the concept of "Precision Medicine" and how this concept is related to articifical intelligence.
2) In the discussion section I believe it could be worth mentionig the role that Artificial Intelligence may play also in the therapeutic choices (see PMID: 33299440 for useful citation and further details)
3) The authors correctly match the results of the 3 different CNNs compared to 2 clinicians: in order to to strengthen the comparison it could be worth to underline the clinical background (specialization), and the years of expertise of the 2 clinicians.
Round 2
Reviewer 1 Report
The authors correctly addressed my concerns in the reviewed version.